# Do You Know What You Eat? Kebab Adulteration in Poland

**DOI:** 10.3390/foods12183380

**Published:** 2023-09-09

**Authors:** Artur Szyłak, Wiktoria Kostrzewa, Jacek Bania, Aleksandra Tabiś

**Affiliations:** Department of Food Hygiene and Consumer Health Protection, Wroclaw University of Environmental and Life Sciences, 50-375 Wroclaw, Poland; 115879@student.upwr.edu.pl (A.S.); 115720@student.upwr.edu.pl (W.K.); jacek.bania@upwr.edu.pl (J.B.)

**Keywords:** food adulteration, kebab, PCR, meat authenticity

## Abstract

In recent years, consumer interest in meat authenticity has increased. Fraudulent claims are most likely to be regarding meat origin, meat substitution, meat processing treatment, and non-meat ingredient additions. This study focuses on the substitution of meat species in processed kebab-like food sales in Poland. The growing popularity of kebab-like foods and the limited number of official inspections of this type of food make this topic interesting. In this study, the results reveal that 60% of the foods analyzed contain an undeclared ingredient or the substitution of an expensive ingredient with a cheaper option.

## 1. Introduction

Food adulteration is a practice that has been known for centuries and, as a result, laws have been established to protect the consumer [1]. The motivations behind food adulteration vary but can include economic factors, the similarity and diversity of animal species, and stock limitation [2]. There is no justification for food fraud, as food manufacturers have a moral and legal obligation not to mislead customers [1]. The shape of European food laws is dynamic, and the horsemeat scandal in 2013, during which horsemeat of unknown origin was found in beef lasagna, was a significant influence. Since then, the framework of European legislation has changed. According to Regulation 178/2002, government and state agencies have an obligation to monitor and enforce legislation on food safety [3]. However, despite the improved standards of food safety in the EU, it is unlikely that future food safety crises can be completely avoided [1]. This is the first study of the adulteration of kebabs in Poland. Interestingly, the first official inspection ever performed at the level of individual kebab sales points in Poland also took place this year. The July 2023 inspection by the Polish Agricultural and Food Quality Inspection, regarding the commercial quality of foods offered in the form of kebabs, showed a number of irregularities. Out of 98 batches of dishes, irregularities were detected in 53 of them. The main complaints were the lack of a declared ingredient and substitution for cheaper meat, or that a declared ingredient was present together with another type of meat. The nationwide inspection results were published on the official website and most of the media spread this information among consumers. Deliberate alteration of food quality is a relatively common practice, but the real scale is difficult to determine due to the heterogeneity of fraud and the high costs of the necessary control methods. Awareness surrounding the human tendency to prioritize profits leads us to question how reliable the declarations of fast food and kebab producers are regarding the meat contents of the processed foods used to produce kebabs [4]. Adulteration of meat products may include reducing the amount of meat in the product or using cheaper substitutes from other animal species instead of the meat advertised [2,4,5,6].

By definition, the meat used in a traditional Iraqi “kebab” is mixed ground beef, mutton, or goat meat [4]. Apart from exposing consumers to economic loss, where more expensive ingredients (lamb, mutton, beef) are replaced with cheaper ones (chicken, pork), food adulteration also has health consequences. Substitution of ingredients other than those declared may cause allergies. Undeclared products of unknown origin may contain residues of antibiotics or other chemical hazards. Ingredients of unknown origin can be a source of biological hazards such as mold or bacteria. The analysis of alerts from the RASFF system (rapid alert system for food) in the period from January 2022 to August 2023 showed that the notifications mainly concern the presence of Salmonella spp. in chicken kebab meat from Poland (out of 37 notifications, 24 are related to this bacteria). Although the bacteria should not be a threat after heat treatment, there is always a risk of becoming sick, which was the case in August of this year in Austria. As a result of ingesting kebab meat contaminated with Salmonella from Poland, a 63-year-old resident of Austria died [7]. Changes in European law after 2019 and the introduction of Regulation 625/2017 made it possible for unqualified persons to exercise official control over foods of animal origin. Unfortunately, this may have consequences in terms of the hygienic quality of the products. Moreover, there is no system of coordinated inspections in Poland between inspectors responsible for the different stages of production. The absence of such a system makes the current controls insufficient and exposes consumers to economic, health, or emotional losses. Controls carried out at each stage of production allow for the elimination of products of unknown origin from the food chain, but the last stage carries the highest risk because it directly threatens consumers. Undeclared meat of unknown origin should be treated as a category 1 material and handled, in accordance with Regulation 1069/2009, as only for disposal [8]. Using a method based on DNA-analysis techniques, we were able to determine in our research whether the declared type of meat is present within the product at all and whether there are any additions of meat from other species of slaughter animals. There are many different methods of meat-species composition testing: the protein-based method, the DNA-based method, spectroscopic methods [9], and chromatography [10]. DNA is relatively stable under common food processing conditions, such as high temperatures, pressures, and chemical treatment [6,11,12], compared to the protein used in protein-based methods [10]. In this study, most frequently, the PCR-based method was used: DNA amplification with species-specific primers.

## 2. Materials and Methods

### 2.1. Samples

A survey of the quality of processed products was conducted on 35 samples of kebabs purchased in 2022. In total, 27 kebab sales points were taken into consideration. Four samples came from Germany and thirty-one samples were collected in Poland. Each specimen was kept at −18 °C until use. To determine the assay detection limit and sensitivity, the meat samples from different species were mixed with turkey meat. Different samples containing from 0.001 to 10% of the meat of a given species were prepared in accordance with Table 1. Meat specimens for mixture preparation were raw, and the percentages were based on wet weight. Samples were weighed on an analytical balance (Radwag, Poland). For each species, all the tests were performed in two replicates. The positive samples consisted of meat samples from sheep (*Ovis aries*), chicken (*Gallus gallus*), cow (*Bos taurus*), and pig (*Sus scrofa*), collected from local grocery stores in Wroclaw and stored at −20 °C until analysis. Samples of meat were taken from several different muscle parts. The pig meat originated from pork shank and pork ham. The cow meat came from beef brisket. The sheep meat was taken from saddle of lamb. The chicken meat was collected from chicken breasts. Every manufacturer claimed a lack of freezing or any other technological product processing. The meat was packed into sterile 50 mL tubes to prevent cross-contamination.

### 2.2. DNA Isolation

Individual fast food samples (1–2 g), positive samples, and the mixture samples were mixed with a TSM buffer (0.2 TRIS-HCl pH8.0, 0.1 M EDTA, 1% SDS) and minced (Tissueruptor, Qiagen, Germany) in a sterile tube. Samples were incubated with 60 μL of proteinase K (5 mg/mL) overnight at 65 °C. According to the manufacturer’s instructions, DNA extraction was performed using a commercial Food-Extract DNA Purification Kit (EURx, Gdańsk, Poland). The concentration of DNA was measured using a DeNovix spectrophotometer (DeNovix Inc., Wilmington, DE, USA). For the PCR reaction, 100 ng of DNA of kebab samples and 10 ng of control were used.

### 2.3. PCR Reaction

The PCR reaction was performed in a mixture of 10× polymerase DreamTaq buffer (Thermo Scientific, Waltham, MA, USA), 20 nmole of each primer (Genomed, Warsaw, Poland; Table 2), 200 μM of each deoxynucleotide triphosphate (Thermo Scientific), 1 U of DreamTaq DNA polymerase (Thermo Scientific), and 0.5 μg of DNA in a final volume of 25 μL. PCRs were carried out in a SimpliAmp Thermal Cycler (Applied Biosystems, Warszawa, Poland) using the following PCR protocol: 1 min at 95 °C followed by 35 cycles of 95 °C for 30 s, 58 °C (for sheep, cow and pig) or 60 °C (for chicken) for 30 s, and 72 °C for 30 s. A final elongation step was performed at 72 °C for 10 min. The results were visualized on a 1.5% agarose gel.

## 3. Results

The limit of detection of our method, determined on 10 points per dilution, was shown to range from 0.1 to 0.0001%, depending on species. Analysis of the amplicons obtained (Figure 1) from the mixtures of turkey containing 10–0.0001% of raw lamb (Figure 1A), chicken (Figure 1C), beef (Figure 1B), and pork meat (Figure 1D) demonstrated that chicken and beef DNA were detected with the highest sensitivity, since a strong signal could be detected even at 0.0001% of these meats in turkey meat. The pork DNA detection rate was 0.001% of the DNA in turkey meat. The weakest level of detection was observed in the DNA of lamb, where the signal was observed at 0.1% lamb in the mix with turkey meat. No cross-reactivity with the other above-mentioned species was observed.

The kebab adulteration analysis was based on the 35 meat samples. Using the PCR method and the confirmed sensitivity to the detection of the tested animal species, we conducted a study in which we proved that only 28.6% (10 samples) of the total samples contained the composition declared by the manufacturer (Table 3). The chicken kebab was the least fraudulent food, nine out of ten samples contained chicken meat as the only ingredient as declared by the seller. An additional 60% of the samples (21 samples) contained meat that was replaced with a cheaper substitute or unlabeled ingredients beyond those declared. It was usually chicken meat (six samples) or a mixture of chicken meat, beef, pork, and lamb (fifteen samples) replacing lamb or beef (Table 3). In 11.4% of the samples (four samples), primers did not detect any of the tested species. An example of a PCR electropherogram can be found in the Appendix A. Lamb/mutton and beef were the most adulterated meat in this study. Of the 17 kebab samples declared to be made of mutton or lamb, only 1 sample was not adulterated (5.9%), and in 2 samples no PCR product was obtained with the primers used. A total of 82.3% of mutton kebab samples were adulterated. Of the 10 samples declared as beef kebab, 9 were adulterated with chicken meat or contained a mix of different types of meat (90%). In one sample, there was no PCR product obtained with the primers used.

## 4. Discussion

According to the OBOP (Public Opinion Research Centre, Warsaw, Poland), kebabs in Poland have become the most popular dish eaten outside the home. Five million Poles eat kebabs every day, and a third of all restaurants in the country serve them [15]. Therefore, it seems reasonable that consumers are concerned about the meat they eat, and accurate labelling is important to maintain transparency. Consumer choice can reflect aspects of health concerns (e.g., absence of allergens) or religion (e.g., absence of pork from some kosher or halal foods) [10]. Meat of unknown origin and meat not declared on the label can cause food poisoning [16]. In addition, accurate labelling is important to support fair trade. Kebab-like foods seem to be particularly vulnerable to fraudulent practices due to their popularity and the fact that there have been no inspections of restaurants and sales points offering Kebab dishes so far. The first official control performed by the Agricultural and Food Quality Inspection in Poland took place in 2023 and showed many irregularities [17]. In the literature, authors focus on the study of an effective method of analyzing the authenticity of kebabs [4,9,18]. Only one study disclosed several systematically deficient regulatory requirements in Döner kebab labelling [19]. In this study, we showed, using the PCR method and confirmed sensitivity to the detection of the tested animal species, that only 28,6% (10 samples) of the samples contained the composition declared by the manufacturer. A total of 60% of the tested samples contained a substitution of a different type of meat than declared or the addition of meat of a different species without marking on the label. The method applied here does not allow the quantitative measure of meat content and it is difficult to say whether the undeclared ingredient is a small addition or the majority of the product. In most adulterated samples, the replacement of a more expensive ingredient (lamb) with a cheaper one (chicken, pork) is observed. Similar results were obtained by the Agricultural and Food Quality Inspection in Poland [17]. The agency’s research showed that the replacement of the declared species with another took place in the case of 54.1% of the tested samples (53 out of 98 samples). In this study, four samples obtained no amplicons among the tested species. An explanation for this may be the use of a different species of meat than the species tested in this study. Unfortunately, this is one of the disadvantages of this method, for which it is necessary to assume in advance what species will be analyzed. Another reason may be the sensitivity of the method used. Of the DNA-based methods, real-time PCR is much more sensitive than classical PCR [20]. A number of PCR-based methods for authentication of common slaughter animal materials in foodstuffs had previously been developed, but there is still a need to expand the possibility of species identification. So far, there is no perfect technique for detecting the substitution of components with components of a different species origin. DNA-based techniques cannot detect adulterated fats or protein, but remain the most universal [10]. Due to the development of techniques based on next-generation sequencing, they are becoming more available and economically justified. 

The main advantages of this technique are the high sensitivity, and the lack of need to set up the composition before performing the analysis (as is the case with techniques based on specific primers). In Poland, future research should focus on the adulteration of fast food, due to the lack of sufficient controls and its high popularity. Very little is known about the authenticity of popular fast foods like pizza or burgers. Official inspections should increase the number of food authenticity controls in restaurants, food trucks, and other points selling food, as happened in Denmark after the horse meat scandal. An interesting way of informing consumers about the quality of meat is the introduction of a blockchain system with encoded information about the quality of the product in accordance with the principle “from farm to table”.

## 5. Conclusions

Despite many years of legislative efforts in the European Union, the problem of food fraud remains relevant. A total of 60% of the kebabs tested in this study were adulterated by the substitution of one species with another. The conducted research can be developed and improved in the future with other parameters, which proves its potential. We can also further consider the meats of other animal species that could be used as cheap substitutes, such as horse meat, and the meat of animals that have entered production through non-compliance with food safety regulations, such as rodents or insects.

## Figures and Tables

**Figure 1 foods-12-03380-f001:**
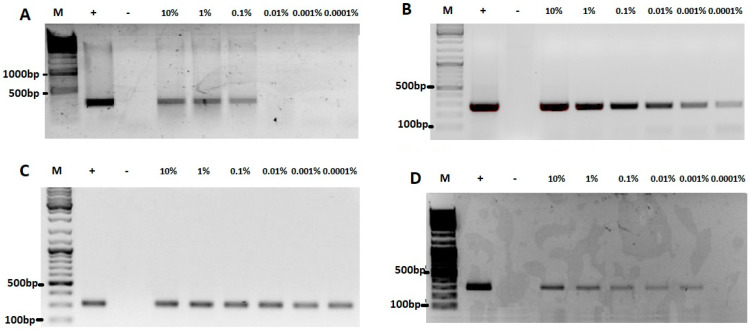
Electrophoretic analysis of amplicons from DNA obtained from 0.0001–10% mixtures of (**A**) lamb, (**B**) beef, (**C**) chicken, and (**D**) pork meat with turkey meat. M—molecular weight marker (Thermo Scientific Gene Ruler DNA Ladder Mix), “+”—100% meat positive control, “−“—no template control.

**Table 1 foods-12-03380-t001:** The detection sensitivity sample preparation.

Percentage of Analyzed Species Meat	Meat of Analyzed Species *	Meat of Non-Analyzed Species
10%	10 g analyzed species meat	90 g of turkey meat
1%	1 g analyzed species meat	99 g of turkey meat
0.1%	0.1 g analyzed species meat	99.9 g of turkey meat
0.01%	0.01 g analyzed species meat	99.99 g of turkey meat
0.001%	0.001 g analyzed species meat	99.99 g of turkey meat
0.0001%	1 g of 0.001% analyzed species meat	99 g of turkey meat

* sheep (*Ovis aries*), chicken (*Gallus gallus*), cow (*Bos taurus*), and pig (*Sus scrofa*).

**Table 2 foods-12-03380-t002:** Primer list.

Species	Primer Sequence (5′ → 3′ End)	Amplicon Size (bp)	Reference
sheep (*Ovis aries*)	TCTGTCTTAAACATGCAAACGA	197	This study
	GTCTATGTTACATTAATAC		
chicken (*Gallus gallus*)	TACCATGTTCTAACCCATTTGG	208	[13]
	AGTTCAGGAGTTATGCATGG		
cow (*Bos taurus*)	CCAATAACTCAACACA	300	[14]
	CGTGATCTAATGGTAAGGAAT		
pig (*Sus scrofa*)	CACGCGCATATAAGCAGGTAA	324	[13]
	CAGATTGTGGGCGTATACT		

**Table 3 foods-12-03380-t003:** Results of PCR kebab analysis.

Sample No.	Declared Meat	Result	Sample No.	Declared Meat	Result
1	Lamb	lamb, chicken	19	lamb	-
2	Chicken	chicken	20	chicken	beef, chicken
3	Lamb	beef, chicken	21	beef, lamb	chicken
4	Chicken	chicken	22	lamb	lamb, chicken
5	Beef	chicken	23	beef, lamb	-
6	Lamb	beef, chicken	24	beef	-
7	Chicken	chicken	25	chicken	chicken
8	Lamb	lamb, beef, chicken,	26	beef	chicken
9	Beef	beef, chicken	27	chicken	chicken
10	Lamb	beef, chicken,	28	chicken	chicken
11	Lamb	beef, chicken, pork	29	lamb	lamb, beef, chicken
12	Lamb	beef, chicken,	30	beef, lamb	chicken
13	Lamb	lamb, beef, chicken	31	beef	chicken
14	Lamb	chicken, pork	32	beef	chicken, beef
15	Chicken	-	33	beef	chicken
16	Chicken	chicken	34	chicken	chicken
17	Chicken	chicken	35	lamb	lamb
18	Lamb	beef, chicken, pork			

## Data Availability

The data presented in this study are available in the published article.

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
