# Peer review of "Do You Know What You Eat? Kebab Adulteration in Poland"

_foods, 2023, doi:10.3390/foods12183380_

Round 1

Reviewer 1 Report

Specific comment:

Line 9-15 : looks like the font size is not uniform

Line 52-53 : How can you explain that the number of your sample can represent Poland?

Line 19-49 : You should state what is the novelty/importance of this communication

General comment:

You should more elaborate on these issues below:

Consumer Awareness: How aware are consumers in Poland about the potential adulteration of kebabs? Have there been any public awareness campaigns or educational initiatives to inform consumers about this issue?

Health Implications: Could the authors elaborate on the potential health risks associated with consuming adulterated kebabs? Are there any documented cases of health issues arising from such adulteration in Poland?

 Regulatory Measures: What regulatory measures are currently in place in Poland to combat food adulteration, specifically for kebabs? Are there any plans to strengthen these regulations in the future?

Economic Implications: How does the adulteration of kebabs impact the economic dynamics of the food industry in Poland? Are consumers paying premium prices for adulterated products?

Technological Solutions: Are there any technological solutions or innovations that can be employed to detect adulteration in kebabs more efficiently and at a larger scale?

 Future Research: What are the authors' recommendations for future research in this area? Are there other popular foods in Poland that might be prone to similar adulteration?

Cultural Implications: How does the adulteration of kebabs, a popular dish, impact the cultural and social dynamics in Poland? Are there any community or religious groups particularly affected by this?

 Comparison with Other Countries: How does the situation in Poland compare to kebab adulteration in other countries? Are there lessons to be learned from how other nations handle this issue

Author Response

Specific comment:

Line 9-15 : looks like the font size is not uniform

Has been improved.

Line 52-53 : How can you explain that the number of your sample can represent Poland?

The number of samples could be larger, which is why the authors decided to publish this study as a short communication. Due to the shocking results, we decided not to wait and publish these data as soon as possible. Comparing our results to other similar studies, this is the average number that researchers publish. As an example: 52 samples were used to identify the species of meat in pet food [1], and only 10 samples were in the study Horvath-Ungerboeck at. al [2], the mislabeling study of fish products in Italy was based on 71 samples [3] and a similar study was made on 26 samples in Los Angeles.

[1]Tara A. Okuma, Rosalee S. Hellberg,Identification of meat species in pet foods using a real-time polymerase chain reaction (PCR) assay,Food Control,Volume 50,2015,9-17,https://doi.org/10.1016/j.foodcont.2014.08.017.

[2] Horvath-Ungerboeck, Christa; Widmann, Karoline; Handl, Stefanie (2017). Detection of DNA from undeclared animal species in commercial elimination diets for dogs using PCR. Veterinary Dermatology, doi:10.1111/vde.12431

 [3] Filonzi, L.; Vaghi, M.; Ardenghi, A.; Rontani, P.M.; Voccia, A.; Nonnis Marzano, F. Efficiency of DNA Mini-Barcoding to Assess Mislabeling in Commercial Fish Products in Italy: An Overview of the Last Decade. Foods 202110, 1449. https://doi.org/10.3390/foods10071449

[4] Willette DA, Simmonds SE, Cheng SH, Esteves S, Kane TL, Nuetzel H, Pilaud N, Rachmawati R, Barber PH. Using DNA barcoding to track seafood mislabeling in Los Angeles restaurants. Conserv Biol. 2017 Oct;31(5):1076-1085. doi: 10.1111/cobi.12888. Epub 2017 May 10. PMID: 28075039.

.

Line 19-49 : You should state what is the novelty/importance of this communication

Has been improved. In a line 28 authors add: This is the first study of the adulteration of kebabs in Poland.

General comment:

You should more elaborate on these issues below:

Consumer Awareness: How aware are consumers in Poland about the potential adulteration of kebabs? Have there been any public awareness campaigns or educational initiatives to inform consumers about this issue?

Line 35:

The nationwide inspection results were published on the official website and most of the media spread this information among consumers.

Health Implications: Could the authors elaborate on the potential health risks associated with consuming adulterated kebabs? Are there any documented cases of health issues arising from such adulteration in Poland?

Added to the introduction, line 45:

Apart from exposing consumers to economic loss, where more expensive ingredients (lamb, mutton, beef) are replaced with cheaper ones (chicken, pork), food adulteration has also health consequences. Substitution of ingredients other than those declared may cause allergies. Undeclared products of unknown origin may contain residues of antibiotics or other chemical hazards. Ingredients of unknown origin can be a source of biological hazards such as mold or bacteria. The analysis of alerts from the RASFF system (rapid alert system for food) in the period from January 2022 to August 2023 showed that the notifications mainly concern the presence of Salmonella spp. in chicken kebab from Poland (out of 37 notifications, 24 are related to this bacteria).  Even though after heat treatment, the bacteria should not be a threat, there is always a risk of getting sick, as was the case in August in Austria. As a result of ingesting contaminated Salmonella Kebab from Poland, a 63-year-old resident of Austria died [1]

 Regulatory Measures: What regulatory measures are currently in place in Poland to combat food adulteration, specifically for kebabs? Are there any plans to strengthen these regulations in the future?

Line 53:

Changes in European law after 2019 and the introduction of Regulation 625/2017 made it possible for unqualified persons to exercise official control over food of animal origin. Unfortunately, this may have consequences in terms of the hygienic quality of the products. Moreover, there is no system of coordinated inspections in Poland between inspections responsible for different stages of production. The absence of such a system makes controls insufficient and exposes consumers to economic, health, or mental losses.

Economic Implications: How does the adulteration of kebabs impact the economic dynamics of the food industry in Poland? Are consumers paying premium prices for adulterated products?

Added to the introduction, line 45:

Apart from exposing consumers to economic loss, where more expensive ingredients (lamb, mutton, beef) are replaced with cheaper ones (chicken, pork), food adulteration has also health consequences.

Technological Solutions: Are there any technological solutions or innovations that can be employed to detect adulteration in kebabs more efficiently and at a larger scale?

Line 180:

So far, there is no perfect technique for detecting the substitution of components with components of a different species origin. DNA-based techniques cannot detect adulterated fats or protein but remain the most universal [2]. Due to the development of techniques based on next-generation sequencing, they are becoming more available and economically justified. The main advantage of this technique is high sensitivity, and no need to set up the composition before performing the analysis (as is the case with techniques based on specific primers).

Line 192:

An interesting way of informing consumers about the quality of meat is the introduction of a blockchain system with encoded information about the quality of the product in accordance with the principle “from farm to table”.

 Future Research: What are the authors' recommendations for future research in this area? Are there other popular foods in Poland that might be prone to similar adulteration?

Line 189:

Very little is known about the authenticity of popular fast foods like pizza or burgers.

Line: 198:

 The conducted research can be developed and improved in the future with other parameters, which proves their potential. We can consider the meat of other animal species that can be a cheap substitute - like horse meat, and meat of animals that have entered through non-compliance with food safety regulations such as rodents or insects

Cultural Implications: How does the adulteration of kebabs, a popular dish, impact the cultural and social dynamics in Poland? Are there any community or religious groups particularly affected by this?

There are no official reports. Since it was the first control, we still have to wait for the results of the decrease in sales.

 Comparison with Other Countries: How does the situation in Poland compare to kebab adulteration in other countries? Are there lessons to be learned from how other nations handle this issue

Line 189:

Official inspections should increase the number of food authenticity controls in restaurants, food trucks and other points selling food, as happened in Denmark after the horse meat scandal.

Reviewer 2 Report

The topic “Do you know what you eat? Kebab adulteration in Poland” is fine in the field, and the results in the present study are interesting to show the adulteration in kebab food. However, some point of the manuscript needs to clarify and revise.

1. Typographical errors exist throughout the manuscript. Rectify carefully.

2. Please mention more detail about the adulteration in kebab in the introduction section.

3. What is the reason behind why researchers divide the Percentage of analyzed species meat by many numbers? Please justify this point.

4. The research article addresses the adulteration in kebab products with any meat species, if they had to enter the biological systems what will be the ways of elimination or the fate of disposal from the biological food chain? Please justify it.

5. In the conclusion seems to be in general and is not given separately, it is highly recommended to include limitations of the study and potential future research goals.

Minor checking and revise in the whole manuscript based on the comment to the authors.

Author Response

  1. Typographical errors exist throughout the manuscript. Rectify carefully.

Has been improved.

  1. Please mention more detail about the adulteration in kebab in the introduction section.

In line 32 added:

Out of 98 batches of dishes, irregularities were detected in 53 of them. The main complaint was the lack of the declared ingredient and substitution for cheaper meat or this ingredient was present together with another type of meat.

  1. What is the reason behind why researchers divide the Percentage of analyzed species meat by many numbers? Please justify this point.

The reason why we divided the percentage of analyzed meat species by many numbers was to test the sensitivity of the method used. Before using primers appropriate for a given species, we checked their sensitivity in previously prepared samples of the tested meat, which was mixed with another species (turkey meat) in the appropriate concentration, confirming the percentage content.

  1. The research article addresses the adulteration in kebab products with any meat species, if they had to enter the biological systems what will be the ways of elimination or the fate of disposal from the biological food chain? Please justify it.

Controls carried out at each stage of production allow for the elimination of products of unknown origin from the food chain, but the last stage carries the highest risk because it threatens directly to consumers. In the case of undeclared meat of unknown origin, it should be treated as category 1 material and handled in accordance with Regulation 1069/2009, only for disposal.

  1. In the conclusion seems to be in general and is not given separately, it is highly recommended to include limitations of the study and potential future research goals.

Line 198:

The conducted research can be developed and extended in the future with other parameters, which proves their potential. We can consider the meat of other animal species that can be a cheap substitute - like horse meat, and meat of animals that have entered through non-compliance with food safety regulations such as rodents or insects

Comments on the Quality of English Language

Minor checking and revise in the whole manuscript based on the comment to the authors.

the text has been checked by a native speaker